# Protein Microarrays for High Throughput Hydrogen/Deuterium Exchange Monitored by FTIR Imaging

**DOI:** 10.3390/ijms25189989

**Published:** 2024-09-16

**Authors:** Joëlle De Meutter, Erik Goormaghtigh

**Affiliations:** Center for Structural Biology and Bioinformatics, Laboratory for the Structure and Function of Biological Membranes, Campus Plaine, Université Libre de Bruxelles CP206/2, B1050 Brussels, Belgium

**Keywords:** FTIR spectroscopy, hydrogen deuterium exchange, protein microarrays

## Abstract

Proteins form the fastest-growing therapeutic class. Due to their intrinsic instability, loss of native structure is common. Structure alteration must be carefully evaluated as structural changes may jeopardize the efficiency and safety of the protein-based drugs. Hydrogen deuterium exchange (HDX) has long been used to evaluate protein structure and dynamics. The rate of exchange constitutes a sensitive marker of the conformational state of the protein and of its stability. It is often monitored by mass spectrometry. Fourier transform infrared (FTIR) spectroscopy is another method with very promising capabilities. Combining protein microarrays with FTIR imaging resulted in high throughput HDX FTIR measurements. BaF_2_ slides bearing the protein microarrays were covered by another slide separated by a spacer, allowing us to flush the cell continuously with a flow of N_2_ gas saturated with ^2^H_2_O. Exchange occurred simultaneously for all proteins and single images covering ca. 96 spots of proteins that could be recorded on-line at selected time points. Each protein spot contained ca. 5 ng protein, and the entire array covered 2.5 × 2.5 mm^2^. Furthermore, HDX could be monitored in real time, and the experiment was therefore not subject to back-exchange problems. Analysis of HDX curves by inverse Laplace transform and by fitting exponential curves indicated that quantitative comparison of the samples is feasible. The paper also demonstrates how the whole process of analysis can be automatized to yield fast analyses.

## 1. Introduction

Proteins are widely used as biotherapeutics (i.e., antibodies, hormones and enzymes). They form the fastest-growing class of therapeutics and can be applied in a large diversity of disorders, such as cancers, autoimmune diseases or metabolic illnesses [1]. In the course of protein production, storage, transport and delivery to the patient, loss of native structure is common. Structure alteration can include changes in the secondary structure or simply a remodeling of the spatial arrangement of these secondary structure elements, resulting in a reorganization of the protein tertiary structure with insignificant change in the secondary structure. Protein aggregation is part of the structural changes that can be observed and is particularly critical for proteins used as therapeutics [2]. In all cases, the therapeutic efficiency and safety can be compromised.

Hydrogen deuterium exchange (HDX) has long been used for the analysis of protein structure and dynamics. The exchange rate of amide protons depends on their accessibility to solvent and stability/dynamics of the H-bonds in which they are involved. The approach consists in monitoring the exchange of the protein backbone amide hydrogens with deuterium, which provides information on both protein dynamics and conformation. The rate of exchange therefore constitutes a sensitive marker of the conformational state of the protein and of its stability. For this reason, HDX has become increasingly important for establishing the level of similarity between a biosimilar and an innovator drug, as reviewed elsewhere [3,4]. The major tool used to monitor the HDX process is currently mass spectrometry (MS) [5,6]. Most MS-based HDX methods involve a rapid proteolytic digestion followed by Liquid Chromatography–Mass Spectrometry (LC/MS) analysis, e.g., [7], with HDX kinetics monitored at the peptide level. At selected time points, the exchange reaction is quenched at a low pH as the HDX rate is minimal at pH 2.0–3.0 and at a temperature close to 0 °C [8]. For instance, the random coil chain of poly-DL-alanine has a half-time of exchange of about 100 min at pH 3.0 and 0 °C. It decreases to less than one second near pH 7.0 [9]. To minimize back exchange of amide backbone, i.e., -N^2^H to -N^1^H, all subsequent procedures (desalting and separations) are performed under the same low pH and temperature conditions. Yet, back-exchange may still average ~20–30% [10,11], blurring the HDX results; however, it can be very significantly reduced by using rapid reversed-phase separation [12] or partially corrected by deep learning approaches [7]. A recent inter-laboratory comparison of the HDX rate for a Fab fragment reported a reproducibility of 9.0 ± 0.9% after back exchange correction. Other approaches, such as nuclear magnetic resonance (NMR) [13] and Raman spectroscopy [14], are interesting alternatives.

Fourier transform infrared (FTIR) spectroscopy is still another method with very promising capabilities. Since the early works [15], hundreds of papers have reported the use of FTIR spectroscopy for monitoring HDX [16]. Even though FTIR spectroscopy does not bring HDX data at the level of particular segments of the protein, it has the advantage of focusing on amide protons only. Furthermore, HDX can be monitored in real time, and the experiment is therefore not subject to back-exchange problems encountered when the exchange reaction has to be stopped by decreasing the pH before subsequent manipulations of the sample required prior to analysis. HDX monitored by FTIR spectroscopy has been reported to be exquisitely sensitive to conformational changes, including those involved in the course of protein activity, for instance in human aromatase after the addition of its specific ligands [17], in P-type ATPases in the course of their catalytic cycle [18,19,20] and in ABC transporters in the presence of different ligands [21,22]. Simultaneously, no significant secondary structure change could be detected, demonstrating the capability of the method to detect tertiary structure changes. The interest for FTIR monitoring of HDX comes on top of the well-documented capability of FTIR spectroscopy to assess protein secondary structure [23,24,25,26,27,28,29] and, simultaneously, protein glycan content [30] or lipid content [31,32]. HDX therefore adds another dimension to the use of FTIR spectroscopy for monitoring conformational changes at the tertiary structure level that may occur in proteins. In addition, 2D-correlation spectroscopy may help assign the exchange to particular protein segments [33,34,35].

In FTIR spectroscopy, the amide ν(C=O) band, called Amide I, is found between 1700 and 1600 cm^−1^. It is only slightly affected by the HDX process through the contribution of the in-plane δ(N–H) to the potential of Amide I. It experiences a 5–10 cm^−1^ shift to lower wavenumbers upon N–H deuteration [25,36,37]. Interestingly, the δ(N–H) band, called Amide II, found between 1600 and 1500 cm^−1^, completely disappears upon exchanging -N^1^H for -N^2^H, while a new band, called amide II’, appears about 100 cm^−1^ below. Monitoring the overall amide proton exchange is therefore as simple as measuring the amide II band decay as a function of time. All protein amide protons are monitored simultaneously.

In the industry in particular, the challenge is to handle the high number of samples from production lines or from studies, including a high number of environmental conditions, such as those generated by thermal or mechanical stress studies, freeze–thaw cycles, shaking, etc. As far as the secondary structure is concerned (and therefore aggregation), Kazarian and colleagues suggested the use of macro ATR–FTIR imaging to monitor 12 samples simultaneously [38,39]. Recently, we proposed a new approach for evaluating protein secondary structure in a high throughput way, combining infrared imaging and protein microarray printing [29]. The introduction of protein microarrays coupled with FTIR imaging makes the experiment particularly convenient. In this work, the BaF_2_ slides bearing the microarray were covered by another slide but separated by a spacer allowing us to flush the cell continuously with a flow of N_2_ gas saturated with ^2^H_2_O. Exchange then occurred simultaneously for all proteins and single images covering ca. 96 spots of proteins that could be recorded on-line at selected time points. The paper demonstrates the feasibility of the process, using 85 different proteins, listed in Appendix A, belonging to the cSP92 protein library [40], a recently built protein library covering as much as possible the secondary structure and higher order structure space. The paper also develops an analysis of the exchange curves.

## 2. Results

In previous works, FTIR imaging of protein microarrays was obtained using a 15× magnification. In such conditions, the area covered by a unit image is only 0.7 × 0.7 mm^2^, corresponding to ca. 6 protein spots. In order to increase the throughput, a 4× magnification was used in the present work. In such condition, the 2.5 × 2.5 mm^2^ image covered more than 96 protein spots. After the flushing of the HDX chamber of the cell with ^2^H_2_O-saturated N_2_ flow started, spectra were recorded every 3 min. A schematic description of the experimental setup appears in Figure 1.

With such a design, the evolution of HDX could be monitored on 96 proteins simultaneously. The simultaneity of the recording of many samples together brings a huge advantage for comparative studies as it eliminates the problem of experiment-to-experiment variations. Images, each one containing 16,384 spectra, were recorded every 3 min during the first hour, then the time spacing was increased (See Appendix A, “Microarray data pre-processing”).

The detailed procedure for processing the images and the spectra is presented in Appendix A. Spectra belonging to proteins were identified by the signal (absorbance at the maximum of the amide I band) to noise (standard deviation between 2000 and 1900 cm^−1^) ratio (SNR) as detailed in Appendix A. For each image, a first background was removed by subtracting the mean spectrum of an area where no protein had been spotted. Grids defining squares containing each spot were then drawn on the image (Figure 2), and a local background found within the grid-defined square surrounding a spot was subtracted from all spectra of that square, resulting in a definite improvement of the quality of the spectra (Appendix A). Once the cube of data had been processed as indicated above, for each square of each grid, the mean spectrum of the spectra passing the SNR threshold was computed and saved (Appendix A). With the spectra properly named, first according to the grid, then according to the grid square and finally according to the time point, spectra alphanumerically sorted naturally loaded in the sequence of the kinetics, protein after protein (Appendix A). An example of the evolution of the spectra as a function of the time of exposure to ^2^H_2_O is presented in Figure 3. Two significantly different proteins have been selected for this example, myoglobin, a well-structured α-helical protein and metallothionein, a mostly unstructured protein. They can be expected to display very different exchange rates. 

It can be observed in Figure 3 that the amide I band near 1650 cm^−1^, arising essentially from the amide ν(C=O), remains largely unchanged upon deuteration, though it experiences a small shift (ca 10 cm^−1^) to lower wavenumbers. In contrast, amide II near 1545 cm^−1^, deriving from δ(N–H) and ν(C–N), tends to disappear completely upon deuteration. While the intensity drop remains modest for myoglobin, a highly structured protein very rich in α-helices, the effect is much more dramatic for metallothionein, an intrinsically disordered protein expected to experience a fast HDX rate. As amide II intensity decays, a new band called amide II’ rises around 1450 cm^−1^. A small decrease in spectrum intensity can also be observed between 1300 and 1200 cm^−1^ assigned to the Amide III band (Figure 3). A quantitative evaluation of the exchange rate is better obtained after integration of the Amide II band (Appendix A). The fraction of the unexchanged amide protons could therefore be computed for each time point of the kinetics. The insets in Figure 3 report the exchange curve for myoglobin and metallothionein. Detailed examples are reported in Appendix A. Analysis of the integration data required a normalization. This was obtained by dividing the area of amide II by the area of amide I, which is supposed to remain constant in the course of HDX. Zero time was set as 100% ^1^H form, and zero amide II area was set as 0% ^1^H form. It can be observed on Figure 3 that the exchange rate is very different for the two proteins. The highly structured myoglobin exchanged at a considerably slower pace than the intrinsically disordered metallothionein, in agreement with the important number of amide hydrogens engaged in the hydrogen bonds stabilizing the helices present in myoglobin.

The inverse Laplace transform (ILT) of the exchange curves is presented in Figure 4 and Appendix A. ILT is a classical means to decompose a set of overlapping exponential decays. The advantage is that is does not require human decision as to the number of components as it is the case for curve fitting (see below). In the absence of any preconceived hypothesis on the number of components present in the exchange curves, the ILT identified three main populations of amide protons exchanging at different rates: ca. 10^4^ min for the slowest one, 50 min for the intermediate and 6 min for the fastest one. Though with small variations depending on the protein, similar observations could be made with the other proteins tested (Appendix A).

The important observation provided by the ILT curves obtained for all proteins is that two to four categories of amide protons characterized by specific time constants are sufficient to represent the exchange curves obtained within the experimental noise. In the vast majority of the cases, three time constants were identified. As curve fitting is easier to handle, in particular for constraining some parameters to remain either constant or within a given range of values, the subsequent analyses were obtained after curve fitting.

Curve fitting was performed with three or four times constants. The initial values were the median values identified by the inverse Laplace transform approach (Appendix A), except for the fast exchange that was set to 5 min. The fitted HDX curves appear in Appendix A for all proteins, and detailed results are reported in Appendix A. Results of the fit for myoglobin and metallothionein appear in the insets of Figure 3. As there was little gain in fitting standard deviation upon adding a fourth time constant (see the difference of standard deviations obtained for three or four time constants, last column of Appendix A), results obtained with three time constants are presented here. In a first run, both the time constants and the proportion of amide protons exchanging with these time constants were adjusted by the least square curve fitting. Numerical data are reported in Appendix A and are graphically represented in Appendix A. At this stage, time constants optimized for each protein are obviously all different. In order to properly compare protein exchange behavior, it is adequate to compare the fraction of the amide ^1^H exchanging with the same time constant. This was achieved by applying a constrained curve fitting, imposing the time constants as the mean of the values found in the first run (Appendix A). The results are summarized in Figure 5. Figure 5 indicates that the fraction of protons belonging to each of the three time constants varies widely. The sum of the three fractions is 100%, i.e., all the amide protons of the protein. The fast exchange occurred in 11.3 min. For metallothionein, the first protein of the list, 88% of the amide protons belong to this class, while there were almost none in protein #51 (amidase) and #79 (leptin). The intermediate category, exchanging in 68.7 min, included 43% of the amide protons for protein #82 (apolipoprotein E3) but almost none for protein #60 (carboxypeptidase). Finally, the slowly exchanging class involved only 7% of the amide proton for protein #1 (metallothionein) but 89% of the amide proton of protein #79 (calmodulin). While it makes sense that with an intrinsically disordered protein, such as metallothionein, exchanged rapidly with almost no slow exchange contribution, in general, the exchange behavior cannot be easily related to the α-helix content. In Figure 5, proteins have indeed been sorted according to α-helix content, and no obvious correlation can be observed. A quantitative correlation analysis revealed that there is no correlation coefficient higher than 0.35 between the fraction of amide protons in one of the three classes and the secondary structure content (Appendix A). This is not unexpected, as the exchange rate of a secondary structure, such as the α-helix or β-sheet structure, depends essentially on their stability, as recently illustrated on a viral capsid [41].

## 3. Discussion

The work presented here demonstrates the feasibility of monitoring the HDX rate on a large series of proteins simultaneously and in real time. The protocol used for processing the data requires some attention on two issues. The first one is the use of amide I as an internal reference. The use of an internal reference is required when using attenuated total reflection FTIR as hydration induces a swelling of the sample layer, which results in an overall intensify decrease. In the present case, data were recorded by FTIR imaging in transmission mode. Yet, here too, it is recommended because some protein spots may slightly spread over the BaF_2_ support upon hydration. Though very limited, it can be sufficient to modify significantly the overall intensity of the spectra as the spreading results in more protein materials present in pixels discarded because of the low signal-to-noise ratio (see the inset of Figure 6). Using amide I as an internal reference solves this problem but induces another potential problem related to the fact the water δ(O–H) vibration also brings some contribution in the amide I area. The consequence is that this water, always present as a shell of hydration in dried samples, will be rapidly exchanged by ^2^H_2_O, and its contribution will disappear from the amide I spectral region. The two effects, the smear of protein spot and exchange of ^1^H_2_O by ^2^H_2_O, contribute to decrease the amide I intensity. The former is adequate to correct for the loss of materials retained for analysis, but the latter would result in an artifactual decrease of the amide I area evaluation. In order to examine the extent of the amide I area drop in the course of the exchange, the ratio between the area of amide I at the end of the experiment after 370 min and the area of amide I at t = 0 min has been computed. A histogram of these ratios is reported on Figure 6 for the 85 proteins used in this study.

It can be observed that most proteins display a ratio around 0.95 (mean value), which can be safely considered as due to the disappearance of the H_2_O contribution. Yet, a few other proteins display significant deviations. This is, for instance, the case for D-amino acid oxidase, which displays a ratio of 0.6. Careful examination of the evolution of the area of amide I indicates it is very stable during the first hour but decreases almost linearly with time after the first hour (not shown). This decrease is due to some spreading of the protein spot, which was particularly significant for this protein. The inset in Figure 6 illustrates the decrease of protein materials retained for the analysis, i.e. passing the SNR threshold as explained in the Appendix A. Such a behavior is not observed for more concentrated samples, as illustrated in the inset for β-lactamase. Using amide I as the internal standard, therefore, appears to be a good practice in the general case. The fraction of amide I loss of intensity due to the exchange of water is probably around 5%, but it must be stressed that it may vary according to buffer composition, protein concentration (i.e., the relative amount of hydrated buffer and salt materials) and could complicate the comparison between a same protein present in different environments.

The second issue to be considered is the setting of 100% deuteration as a zero area for amide II. In fact, the amide II band is overlapped by some amino acid side chain contributions [18,42,43,44,45,46,47,48,49,50]. After full deuteration, these contributions will remain largely unchanged, resulting in a residual absorbance that depends on amino acid composition. In order to evaluate the impact of the side chains in the present study, we rebuilt the contributions of the amides (according to secondary structure content), and of the amino acid side chains for the 85 proteins, before and after full deuteration. The parameters to build the amino acid side chain contributions for deuterated/undeuterated forms and for low and high pH when relevant, have been summarized in [44], including a MatLab code to produce the curves. We then computed the drop of the amide I/amide II ratio computed as described in the Appendix A upon full deuteration. In 86% of the cases, the area remaining in the amide II band after full deuteration is less than 5% of the area measured before deuteration (the overall average is 3.2%). Yet, for albumin, the remaining area reached 31%. This protein also has the highest content in Asp + Glu, which together accounts for 17% of the amino acid residues, i.e., about 50% more than the mean value. The 9 proteins that had a residual amide II area between 10 and 22% of the original area all had very high Asp + Glu content. This highlights the fact that the assumption of a zero area in the amide III band after deuteration can be problematic for some proteins and bias the comparison among proteins. Examination of the content in Glu and Asp could draw the attention on this issue.

Once these issues have been considered, the paper suggests a procedure to efficiently analyze the data. HDX curves of 85 different proteins have been recorded by exposing solid spots of proteins containing ca. 5 ng protein to a N_2_ atmosphere saturated with ^2^H_2_O. One microarray formed by ca. 100 protein samples and covering an area of ca. 2.5 × 2.5 µm^2^ can be monitored on-line as the exchange proceeds, precluding any difficulty related to back-exchange. ILT was allowed to obtain an unsupervised overview of the populations of amide protons, including the major time constants. This approach was found to be useful for a first exploration of the data but did not allow a quantitative comparison among proteins. Curve fitting is supervised in the sense that the number of time constants must be introduced from the beginning. Based on the ILT results and on the comparison between the fitting standard deviations obtained for three or four time constants, it was decided to use three exponentials throughout. Such a decision is not particularly compelling when studying unrelated proteins but would be adequate for comparing similar samples, such as the same protein that has been exposed to different environments. It was introduced here to illustrate how a quantitative comparison of the fraction of amide protons present in the three classes of exchange rates can be obtained. For such a comparison, the approach is required as it is indeed difficult to draw conclusions when both the time constants and the proportion of amide protons are allowed to vary.

No particular trend emerged concerning a relationship between the rates of exchange and the secondary structure content. It has been observed for a long time that secondary structures or their environments are insufficient to predict HDX behavior. For instance among membrane embedded proteins, HDX of bacteriorhodopsin [51] or rhodopsin [52] is extremely slow for a large fraction (70–80%) of the amide protons, while in the human erythrocyte glucose exchanger, another transmembrane helical protein, more than 80% of the amide protons are exchanged within an hour. Similar results are found with CHIP25 membrane protein [53]. The presence of a large aqueous channel in the protein is not sufficient to ensure fast exchange. Porin from *E. coli*’s outer membrane possesses a large channel, but its exchange kinetics is similar to that of rhodopsin [54]. Similarly, α-helices from pulmonary surfactant SP-C [55] exchange very slowly, while helices in cytochrome c display a very rapid exchange [56]. Furthermore, slow HDX may be observed on the surface of proteins and fast exchange for buried hydrogen [57]. These examples demonstrate that the intrinsic stability of the structures rather than the structures themselves or their environment play a dominant role. This can be rationalized as transient local unfolding that is necessary for the exchange to take place [58,59,60,61]. Importantly, the examples reported here and others [62] also indicate FTIR-monitored HDX works as well for membrane proteins embedded in a lipid membrane. Even though the present paper addressed the overall exchange only, it must be noted that refined analyses using 2D correlation spectroscopy have been developed and could shed light on specific fragments of sequence whose HDX rate is modified [48,63,64,65].

It is legitimate to question how HDX measurements obtained on dried protein spots perform with respect to HDX taking place in solution. As reported by Englander and Kallenbach [59], HDX of lysozyme in a dry powder condition is slower than in solution, but the exchange rate becomes equal to the rate found in solution when the hydration level of the protein exceeds 0.4 g water per gram protein, i.e., when enough water is present to form a monolayer of water around the protein. It was also reported elsewhere that the rate of exchange reaches a stable value when 0.15 g of H_2_O/g lysozyme is present [66]. In conclusion, dried samples exposed to an atmosphere saturated in ^2^H_2_O are expected to have sufficient hydration [50,67] to display a behavior very similar to the behavior found in solution. It must be added that solid state HDX was monitored by FTIR spectroscopy on human serum albumin formulated with sucrose or trehalose at different temperatures. It was able to predict the stability of the protein and the formation of aggregates [68].

In conclusion, protein microarrays open the door to high throughput recording of HDX curves. The approach is robust as many protein samples can be compared in exactly the same conditions, it is devoid of back exchange issues and the processing of the data can be automatized to result in quick quantitative comparison between samples.

## 4. Materials and Methods

### 4.1. Proteins

The proteins belong to cSP92, a protein library recently designed for spectroscopic calibrations [40]. Proteins in cSP92 cover as well as possible the secondary and higher order structure space. Protein samples were solubilized to a final concentration of 10–20 mg/mL in 4 mM Hepes, 85 mM NaCl. Buffer solutions were filtered on 0.2 μm filters before use. To avoid contributions of the original buffers, salts and/or additives of preparation or purification, samples were de-salted and buffer-exchanged against 4 mM Hepes, 85 mM NaCl (5%) pH between 7.4 and 7.6; if not otherwise specified, see [40]. Chemicals and filters are from Merck Life Science, Hoeilaart, Belgium.

### 4.2. Protein Microarray Printing and Imaging

Protein microarray printing and imaging have been described in detail before [29,69,70,71], and the experimental procedure has not been modified for the present work. Spectra were recorded as the average of 64 scans, between 3650 and 900 cm^−1^ at a nominal resolution of 8 cm^−1^. FTIR data were collected in transmission mode using an Agilent mid-IR imager equipped with a liquid nitrogen cooled 128 × 128 Mercury Cadmium Telluride (MCT) Focal Plane Array (FPA) detector and a 4× objective. Each pixel covers an area of 19.5 × 19.5 µm^2^. Automated spectrum extraction is described in detail in the Appendix A, including the procedure followed to subtract the background. After correction for background, spectra filtered for signal-to-noise ratio and maximum absorbance were averaged. Finally, the mean spectra of quadruplicates obtained for a same protein were averaged, yielding one spectrum per protein. Spectra were then baseline corrected by subtraction of a straight line interpolated between the spectral points at 1720 and 1480 cm^−1^. All details are provided in the Appendix A.

### 4.3. Hydrogen Deuterium Exchange

A sealed chamber formed by two BaF_2_ slides separated by a spacer was used to control the atmosphere in contact with the microarray (Figure 1). For the spacer, UHU^®^ Patafix adhesive paste (Bolton Adhesives, Zaventem, Belgium) was used to hermetically seal the chamber interior, including two pieces of catheter ensuring the entry and exit of ^2^H_2_O-saturated nitrogen flow. Once the vessel and catheters were set up and positioned under the microscope, the focus was adjusted, and the device remained in place for the whole duration of the exchange experiment (about 24 h). The use of a 4× objective allowed recording of an entire microarray in one single image. Indeed, a microarray with 96 spots covered an area of about 4.6 mm^2^; the area of a 4× magnification infrared image covered 6.3 mm^2^. IR spectra of the microarrays were recorded as a series of about 30 FTIR images (16,384 spectra each) recorded as a function of the time of exposure to the ^2^H_2_O vapor. The first image, the zero time, was recorded just before connecting the ^2^H_2_O-saturated N_2_ flow to the microarray cell. Nitrogen was bubbling in three vials (assembled in series) at a flow rate of ca. 80 mL N_2_ gas/min controlled by a flow tube (Fisher Bioblock Scientific, Illkirch, France). Bubbling was started at least one hour before starting the experiment. For each image, 64 scans were recorded, which takes about 2 min. About one additional minute was needed to transfer the data to the computer and get the spectrometer ready for the next measurement. The recording time reported here is the average between the beginning of the scanning and the end. During the first hour, images were recorded continuously, i.e., one image every 3 min. After the first hour, the time interval between the two measurements was increased. An example of time series is provided in Appendix A. Appendix A also details the procedure followed to obtain the exchange curves. The supplementary figures presented in Appendix A are numbered S1, S2, …. Briefly, the raw image (Appendix A) is analyzed for the signal-to-noise ratio (SNR) (Appendix A), which is used to select the better spectra (Appendix A). A major advantage of the analysis of the microarray is that the background to be subtracted is recorded on the same microarray and exactly at the same time as the sample. While the subtraction of a general background considerably improved the quality of the data (Appendix A), subtraction of a background located in the immediate vicinity of a protein spot was even more preferable (Appendix A). This is easily achieved after the application of a grid defining an area around each protein spot (Appendix A). 

Once the cube of data had been processed as indicated above, for each square of each grid, the mean spectrum of the spectra passing the SNR threshold was computed and saved (Appendix A). As there are about 30 time points in an HDX kinetic experiment and 96 spots on each grid, 2880 mean spectra were generated for each microarray. In addition, the software (version 2021) also generated mean spectra for each row of each grid (experiments are quadruplicated along the rows), adding 720 mean spectra to the previous series. The methods used to handle these data are described on page S6 of the Appendix A and illustrated in Appendix A.

### 4.4. Analysis of the Kinetics

For any amide proton *i*, the time dependent evolution of the absorbance *A_i_*(*t*) in the amide II spectral range is characterized by a time constant *k_i_*. For a first order reaction, the time evolution of the Amide II absorbance of undeuterated amide proton is given as
(1)$$A_{i}\left(t\right)=A_{i}^{0}\exp\left(-k_{i}t\right),$$
where *A_i_*^0^ is the absorbance of Amide II due to proton *i* before starting the deuteration process. The overall exchange for *N* different amide protons is therefore described as
(2)$$A\left(t\right)=\sum_{i=1}^{N}A_{i}^{0}\;exp(-k_{i}t).$$

Yet, it is not realistic to attempt to identify the time constant for every amide proton of a protein, and, in general, the overall exchange process is described as the sum of a limited number of *M* exponentials, with each one grouping a range of actual time constants: (3)$$A\left(t\right)=\sum_{j=1}^{M}A_{j}\exp\left(-k_{j}t\right),$$
where *A_j_* is the absorbance of all protons belonging to class *j*. Equation (3) expresses that the time-dependence behavior of the spectral variations can be described as the superimposition of a limited number of first-order kinetic processes. Considering a continuous distribution of the time constants *f*(*k*) instead of a discrete distribution *A_j_*, Equation (3) is analogous to the Laplace transform of *f*(*k*):(4)$$A\left(t\right)=\int_{0}^{+\infty }\;f\left(k\right)\exp\left(-kt\right)dk.$$

The inverse Laplace transform (ILT) L^−1^ immediately yields the shape of the distribution of the time constants [48]:(5)$$f(k)=L^{-1}\left\{A\right.\left.\left(t\right)\right\}.$$

Knox and Rosenberg [72] suggested a dimensionless presentation of the distribution function obtained after rewriting of the integral expression *A*(*t*) as follows:(6)$$A\left(t\right)=\int_{0}^{+\infty }k\;f\left(k\right)\exp\left(-kt\right)d\;ln(k).$$

For reasons detailed by Gregory and Lumry [73], the numerical approach to inverse Laplace transform is subject to several artifacts if not carefully treated. We used the program CONTIN [74], kindly provided by Dr. Provencher [74,75], which provides an efficient regularized solution to the problem. It must be noted that the absorbance can be easily converted into the fraction of undeuterated amide protons when considering the area of amide II before deuteration as 100% undeuterated and a zero area of amide II as 100% deuterated (see Appendix A).

The analysis of the evolution of the exchange was started by the integration of Amide I, Amide II and amide II′ bands (Appendix A). Straight curve fitting (Appendix A) or inverse Laplace transform (ILP) (Appendix A) generated time constants and proportions of amide protons related to each time constant. As experiments were carried out in quadruplicates, standard deviations could easily be calculated for each time point of the kinetic (Appendix A).

## Figures and Tables

**Figure 1 ijms-25-09989-f001:**
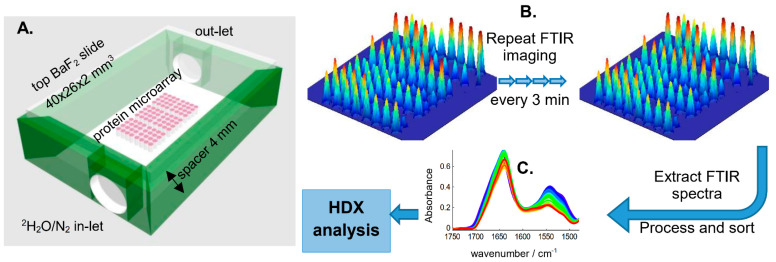
A. Schematic description of the device used for recording HDX kinetics. The cell was built from a bottom 40 × 26 × 2 mm^3^ BaF_2_ slide on which microarrays were deposited. Protein solutions were printed as ca. 100 pl droplets containing each ca. 10 ng protein. On top of a spacer, another BaF_2_ slide sealed the exchange chamber. N_2_ bubbling in ^2^H_2_O arrived through a tubing on one side and was driven out of the microscope-purged chamber by a tubing plugged on the other side. N_2_ bubbled in 3 vials containing ^2^H_2_O placed in series at a flow rate of 80 mL/min. The scale for drawing the microarray in (**A**) has been increased for the sake of visibility. In reality, the entire 96-spot microarray occupied an area of 2.5 × 2.5 mm^2^. (**B**) FTIR images each containing 16,384 spectra were recorded every 3 min at the beginning of the kinetics; time spacing was increased after 1 h. Images in (**B**) report absorbance at 1654 cm^−1^. (**C**) Mean spectra were collected for each spot and each deuteration time, as detailed in Appendix A. They were finally analyzed in terms of HDX kinetics.

**Figure 2 ijms-25-09989-f002:**
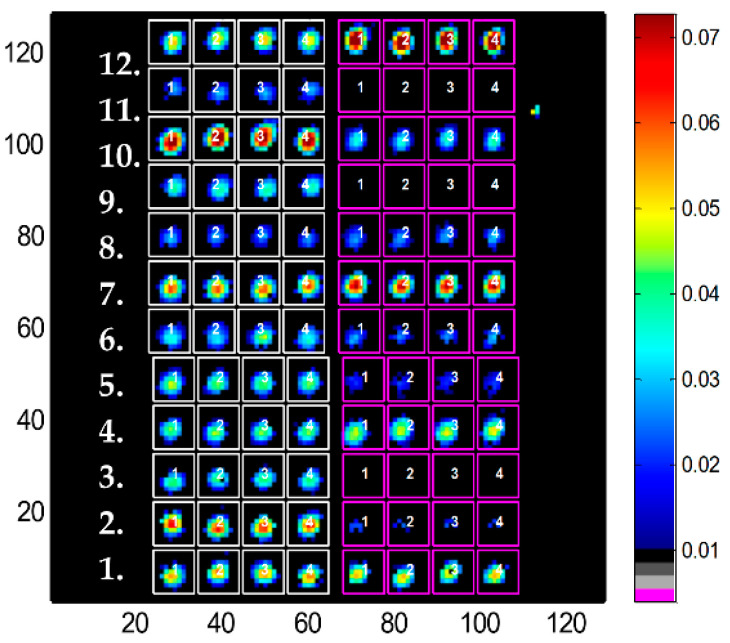
Absorbance at 1654 cm^−1^ of a protein microarray containing 96 protein spots. Pixels with SNR < 35 have been colored in black. Numbers in the left margin identify the 12 proteins of grid #1, drawn in white. Each protein was quadruplicated, resulting in 4 columns. Another grid drawn in magenta identifies another series of 12 samples. Absorbance is color coded, as indicated by the color bar.

**Figure 3 ijms-25-09989-f003:**
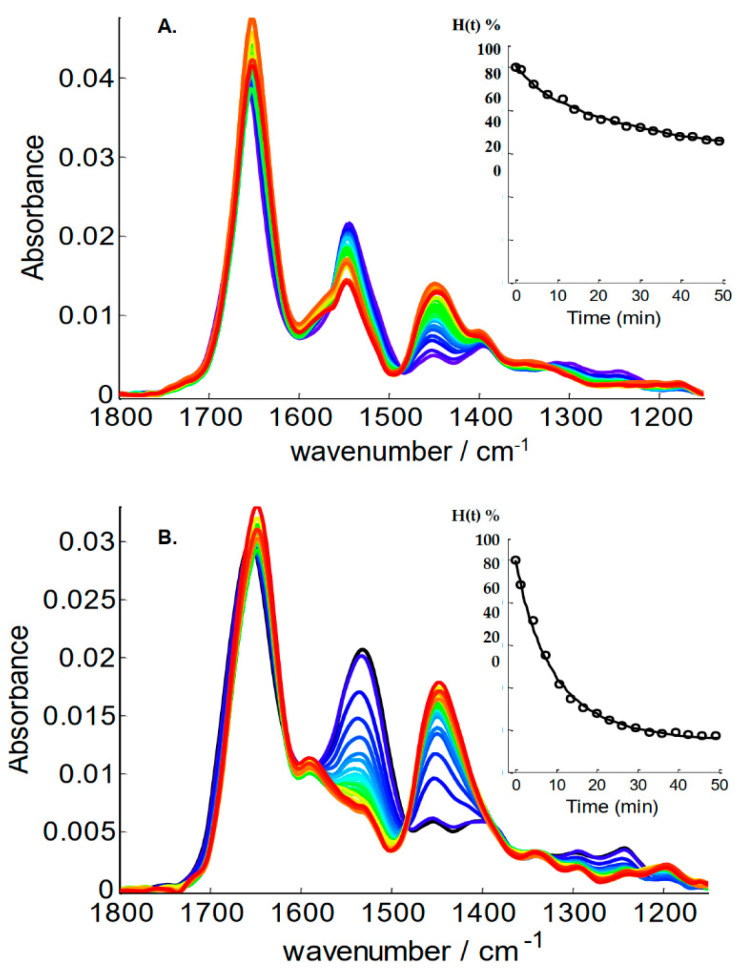
FTIR spectra of (**A**) myoglobin and (**B**) metallothionein in the course of HDX. Times go from 0 (blue spectra) to 1400 min (red spectra). Inset: evolution of the fraction of unexchanged amide protons in the course of the first 50 min; data fitted by 3 exponentials.

**Figure 4 ijms-25-09989-f004:**
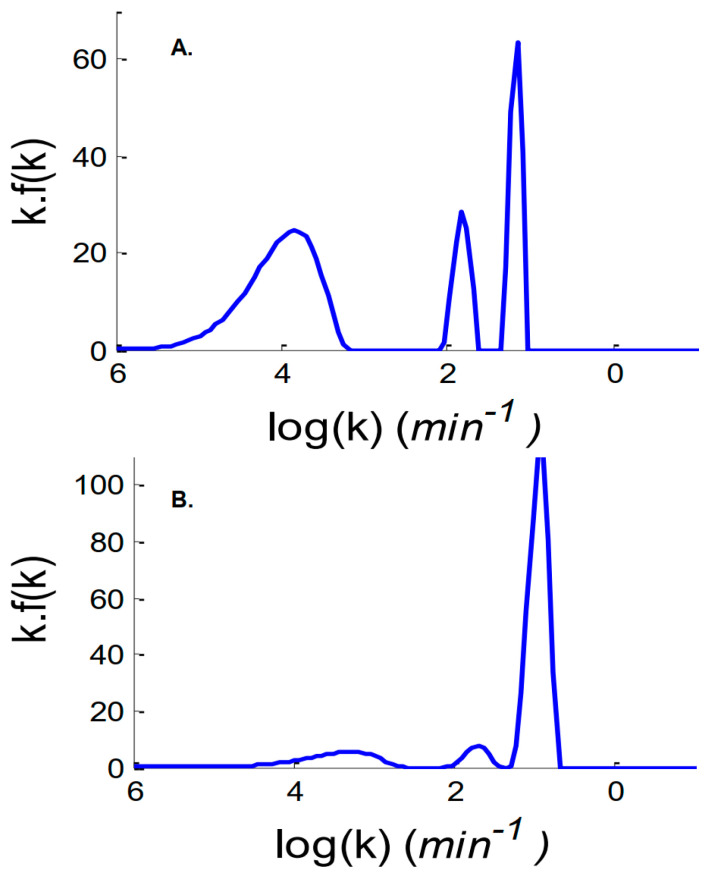
Inverse Laplace transform of the exchange curves for (**A**) myoglobin and (**B**) metallothionein (see Equation (6) in Section 4).

**Figure 5 ijms-25-09989-f005:**
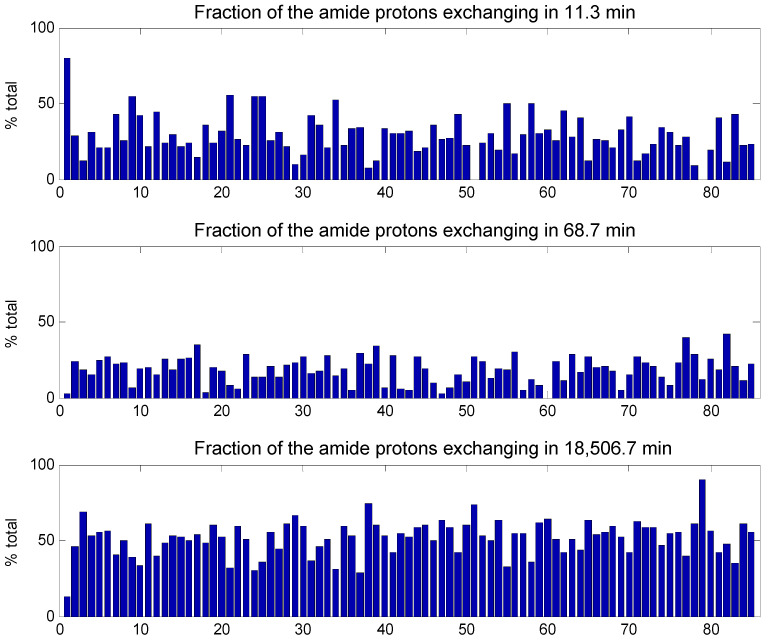
Fraction of the amide protons (%) belonging to 3 classes of exchange rates characterized by time constants of 11.3 min, 68.7 min and 18,506 min. Proteins have been sorted in order of increasing α-helix content, i.e., the order presented in Appendix A.

**Figure 6 ijms-25-09989-f006:**
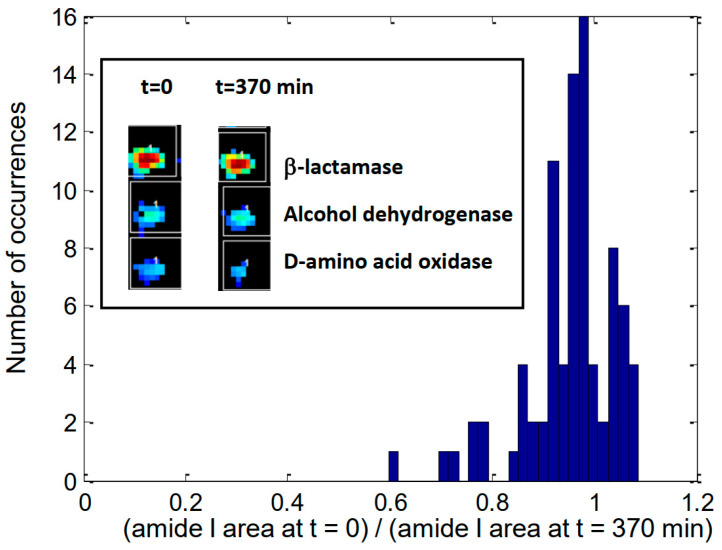
Histogram of the ratio between the area of amide I after 370 min deuteration and its area at t = 0 min for 85 proteins. Inset: picture of 3 protein spots at t = 0 (left column) and y = 370 min (right column) deuteration. Refer to Figure 2 for more details on the image.

## Data Availability

Data are available from the authors from any reasonable request.

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
