# Peer review of "Protein Microarrays for High Throughput Hydrogen/Deuterium Exchange Monitored by FTIR Imaging"

_ijms, 2024, doi:10.3390/ijms25189989_

Round 1

Reviewer 1 Report

Comments and Suggestions for Authors

This article is very well-written and consice. I identified some minor issues that could add value for a broad audience.

On page 2. Replace mass spectroscopy for mass spectrometry

Figure 1: letter on the arrows, put ten below or above.

Please specific what is cSP92 protein library ( or add, refer to methods)

Figure 2. Is the absorbance color code at a specific wavelength? Which one? Otherwise, explain.

On page 4, line 166--167: “An example of the evolution of the spectra as a function of the time of exposure to 2H2O is presented in Figure 3.” I found nice the experiments with these two proteins (controls?) but they are about of the blue. Please explain that they are used as a control or as an example of differential exchange for globular and intrinsically disordered proteins. If this is not the case, explain why you reported these two proteins.

On page 5. Please explain why you employ the Inverse Laplace Transform (ILT) to deconvolute your kinetic information in the text. Perhaps add: “refer to methods”.

 Figure 5. Fraction of the amide protons: Please specify which amide proton.

Reviewer 2 Report

Comments and Suggestions for Authors

This is a report of using FTIR to monitoring HDX for protein dynamics. While not allowing labeling at specific sites, the technique has some advantage (e.g. not subject to back-exchange problem). Two minor issues:

1) Provide more data beyond text descriptions on technique advancement achieved in this study.

2) In Fig. 4, Fraction of amide protons exchanging is much smaller in 68.7 min than in 11.3 min. This does not make any sense. These two sub-figures might be in wrong order.

Comments on the Quality of English Language

There is room for improvement of language.
